

# Determining virus-host interactions and glycerol metabolism profiles in geographically diverse solar salterns with metagenomics

Abraham G. Moller and Chun Liang

Department of Biology, Miami University, Oxford, OH, United States

## ABSTRACT

Solar salterns are excellent model ecosystems for studying virus-microbial interactions because of their low microbial diversity, environmental stability, and high viral density. By using the power of CRISPR spacers to link viruses to their prokaryotic hosts, we explored virus-host interactions in geographically diverse salterns. Using taxonomic profiling, we identified hosts such as archaeal *Haloquadratum*, *Halorubrum*, and *Haloarcula* and bacterial *Salinibacter*, and we found that community composition related to not only salinity but also local environmental dynamics. Characterizing glycerol metabolism genes in these metagenomes suggested *Halorubrum* and *Haloquadratum* possess most dihydroxyacetone kinase genes while *Salinibacter* possesses most glycerol-3-phosphate dehydrogenase genes. Using two different methods, we detected fewer CRISPR spacers in *Haloquadratum*-dominated compared with Halobacteriaceae-dominated saltern metagenomes. After CRISPR detection, spacers were aligned against haloviral genomes to map virus to host. While most alignments for each saltern metagenome linked viruses to *Haloquadratum walsbyi*, there were also alignments indicating interactions with the low abundance taxa *Haloarcula* and *Haloferax*. Further examination of the dinucleotide and trinucleotide usage differences between paired viruses and their hosts confirmed viruses and hosts had similar nucleotide usage signatures. Detection of *cas* genes in the salterns supported the possibility of CRISPR activity. Taken together, our studies suggest similar virus-host interactions exist in different solar salterns and that the glycerol metabolism gene dihydroxyacetone kinase is associated with *Haloquadratum* and *Halorubrum*.

Corresponding author
Chun Liang, liangc@miamioh.edu

## INTRODUCTION

Viruses, the most abundant entities on the planet, are critical for maintaining global biogeochemical cycles (*Suttle, 2007*; *Rohwer, Prangishvili & Lindell, 2009*). Many viruses infect and lyse microbial hosts, releasing stored nutrients back into the ecosystem through a mechanism called the viral shunt (*Suttle, 2007*; *Weitz & Wilhelm, 2012*). This relationship suggests viral-microbial interactions are important for environmental nutrient cycling. Clustered regularly interspaced short palindromic repeat (CRISPR) arrays, which are

found in many archaeal and bacterial genomes (*Grissa, Vergnaud & Pourcel, 2007*), can help us better understand the virus-host interactions that mediate nutrient cycling. Acting as an adaptive immune system against viral infection, CRISPRs incorporate short spacers cleaved from viral DNA into host genomes, providing a record of past infections and thus associations between viruses and prokaryotic hosts (*Sorek, Kunin & Hugenholtz, 2008*).

Virus-host interactions are particularly important in solar saltern ecosystems. These salt-collecting ponds are hypersaline ecosystems that vary from 10 up to 37% (saturation) salt concentration by weight (*Ventosa et al., 2014*). The salterns at Santa Pola near Alicante, Spain contain a low diversity community of halophile microbial hosts, making it a good model ecosystem (*Ventosa et al., 2014*). The square archaeon *Haloquadratum walsbyi* and bacterioidete *Salinibacter ruber* are the two major components of saltern microbial communities at the highest salinities (*Ventosa et al., 2015*). In addition, hypersaline environments are estimated to have the highest densities of viruses (109/mL) among studied aquatic environments (*Santos et al., 2012*). Reduced viral decomposition in hypersaline conditions and a lack of protozoan predation may explain this extraordinarily high density (*Santos et al., 2012*). Though heterotrophic protozoans have been identified in Korean salterns (*Park & Simpson, 2015*), recent studies suggest these species primarily graze upon algae, not bacteria (*Heidelberg et al., 2013*). Although protozoa probably do not engulf bacteria, viruses may indeed prey upon halophilic bacteria and archaea.

While saltern microbial eukaryotes may not be predators of archaea and bacteria, they do serve as the primary producers in these ecosystems. The halotolerant, unicellular green algae *Dunaliella salina* synthesizes vast quantities of glycerol as an osmoprotectant in response to hyperosmotic shock (*Chitlaru & Pick, 1991*). Glycerol that leaks out of these algae is the primary carbon source for the bacteria and archaea that inhabit these ecosystems (*Wegmann, Ben-Amotz & Avron, 1980*; *Oren, 1993*; *Bardavid, Khristo & Oren, 2006*). The rare saltern archaeon *Haloferax volcanii* converts glycerol to glycerol-3-phosphate and then dihydroxyacetone phosphate (DHAP) using glycerol kinase and glycerol-3-phosphate dehydrogenase, respectively (*Sherwood, Cano & Maupin-Furlow, 2009*; *Rawls, Martin & Maupin-Furlow, 2011*). The bacterium *Salinibacter ruber*, on the other hand, oxidizes glycerol to dihydroxyacetone (*Sher et al., 2004*; *Bardavid & Oren, 2007*). Both *Haloferax* and *Haloquadratum walsbyi* can convert dihydroxyacetone to glycolysis intermediate DHAP using dihydroxyacetone kinase (*Bolhuis et al., 2006*; *Bardavid & Oren, 2007*; *Ouellette, Makkay & Papke, 2013*). Considering this metabolic context, haloviruses that lyse halophiles may promote carbon cycling as a viral shunt either by turning over glycerol metabolism intermediates or reducing rates of glycerol uptake within the ecosystem.

In recent years, metagenomic sequencing has made it possible to study microbial community DNA from solar salterns across the planet. Microbial DNA from the Santa Pola salterns has recently been sampled and sequenced (*Fernandez et al., 2013*), in addition to microbial DNA from salterns in Isla Cristina, Spain (*Fernández et al., 2014b*); Cahuil, Chile (*Plominsky et al., 2014*); and Chula Vista, California (*Rodriguez-Brito et al., 2010*). Metagenomics has also made it possible to study the vast haloviral populations that inhabit these ecosystems. A number of viral genomes from the CR30 (**cr**ystallizer, **30**% salt) saltern in Santa Pola have recently been isolated and sequenced using fosmid cloning (*Santos et*

*al., 2012*). Many of these fosmid clones failed to hybridize with microbial DNA, suggesting most of isolated CR30 viruses are lytic (*Santos et al., 2010*). Haloviral genomes have also been assembled from viral metagenomes sequenced from Lake Tyrell in Australia (*Emerson et al., 2013*) and the Chula Vista salterns in California (*Rodriguez-Brito et al., 2010*).

While metagenomic sequencing has provided extraordinary glimpses into the viral and microbial diversity of hypersaline ecosystems, it has also enabled detection of microbial CRISPR arrays. Recent studies in a variety of ecosystems have exploited CRISPRs to link environmental viruses to their microbial hosts (*Tyson & Banfield, 2008*; *Anderson, Brazelton & Baross, 2011*; *Berg Miller et al., 2012*; *Emerson et al., 2013*; *Zhang et al., 2013*; *Sanguino et al., 2015*; *Tschitschko et al., 2015*). Among these, one study examined virus-archaea interactions in an ecosystem similar to salterns—the hypersaline Lake Tyrrell, Australia (*Emerson et al., 2013*). Although this study showed that CRISPRs in the lake were remarkably dynamic over time, it also indicated a number of viruses could be mapped to saltern microbes with CRISPR spacers (*Emerson et al., 2013*). In the saltern context, the abundant community member *Haloquadratum walsbyi* encodes two clusters of *cas* genes (*Bolhuis et al., 2006*; *Dyall-Smith et al., 2011*) and contains CRISPR spacers that map to many Santa Pola haloviral genomes (*Garcia-Heredia et al., 2012*). These studies suggest CRISPRs have great potential to indicate virus-archaea interactions within saltern ecosystems.

In this study, we analyze saltern metagenomes to link haloviruses to their hosts and improve our understanding of carbon cycling amongst the prokaryotes that live in these environments. Furthermore, we incorporate knowledge gained from studying not one but many hypersaline ecosystems across the planet. We not only use metagenomic CRISPR sequences to map virus-host interactions across different saltern microbial populations, but also use relevant taxonomic and functional profiling tools to determine relative taxonomic abundances and relationships between taxa and glycerol metabolism functions. In this context, we find common sets of virus-host interactions and glycerol metabolism gene associations with particular taxa across these geographically distinct salterns. We can also begin to relate lytic infection to specific glycerol metabolism pathways amongst saltern microbes, thus shedding light on the viral contribution to glycerol turnover and thus carbon cycling in solar salterns.

## MATERIALS AND METHODS

### Obtaining metagenomic data sets and haloviral genomes

The Santa Pola, Isla Cristina, and Cahuil microbial metagenomes were downloaded from the NCBI Sequence Read Archive (SRA). Technical details about these microbial metagenomes are provided in Table 1. Further details about sampling are described in associated publications (*Ghai et al., 2011*; *Fernandez et al., 2013*; *Fernández et al., 2014b*; *Fernández et al., 2014c*; *Plominsky et al., 2014*). Chula Vista microbial metagenomes were obtained from the iMicrobe Collaborative (*Hurwitz, 2014*) and have been previously examined in a larger study of several aquatic biomes (*Rodriguez-Brito et al., 2010*). Chula Vista metagenomes were grouped by corresponding salt concentration (low, medium,

**Table 1** **Additional technical information about the Santa Pola, Isla Cristina, and Cahuil solar saltern metagenomes.** This information is provided in the NCBI Sequence Read Archive (SRA) and associated genome announcements (*Fernandez et al., 2013*; *Fernández et al., 2014b*; *Plominsky et al., 2014*). All metagenomes were sequenced using 454 technology. Site names (as listed in the Figures) are listed in bold. NCBI SRA accession numbers (SRX for experiment and SRR for run) are included in the table. Combinations of these metagenomes used for further analyses are listed in Table 3.

| Site | Salinity | Reference | SRX number | Sequencing technology | Number of reads (millions) | Average read length (bp) |
|------|----------|-----------|------------|----------------------|---------------------------|--------------------------|
| **SS13** | 13% | *Fernandez et al. (2013)* | SRX328504 | 454 | 1.5 | 479 |
| **SS19** | 19% | *Ghai et al. (2011)* | SRX024859 | 454 | 0.293 | 504 |
| **IC21** | 21% | *Fernández et al. (2014b)* | SRX352042 | 454 | 1.2 | 508 |
| **SS33** | 33% | *Fernandez et al. (2013)* | SRX347883 | 454 | 0.971 | 429 |
| **SS37** | 37% | *Ghai et al. (2011)* | SRX090229 | 454 | 0.741 | 511 |
| Cahuil (**C34**) | 34% | *Plominsky et al. (2014)* | SRX680116 | 454 | 0.222 | 523 |

**Table 2** **Additional information about the Chula Vista saltern microbial metagenomes obtained from the iMicrobe Collaborative and combined into three metagenomes based on salt concentration.** Site names (as listed in the Figures) are in bold. The datasets can be found at http://data.imicrobe.us/project/view/58.

| Salt content (and label in figures in bold) | Sampling dates corresponding to included metagenomes | Sequencing technology | Number of reads (millions) | Average read length (bp) | Total length (Mbp) |
|---------------------------------------------|------------------------------------------------------|----------------------|---------------------------|--------------------------|--------------------|
| Low (6–8%) (**CV6-8**) | 07/2004 11/10/2005 11/28/2005 | 454 | 0.352 | 95 | 33.4 |
| Medium (12–14%) (**CV12-14**) | 11/10/2005 11/11/2005 11/16/2005 11/28/2005 | 454 | 0.223 | 99 | 22.1 |
| High (27–30%) (**CV27-30**) | 07/2004 11/28/2005 | 454 | 0.618 | 102 | 63.2 |

or high) and concatenated to form single low (CV6-8), medium (CV12-14), and high (CV27-30) salt concentration metagenomes, as shown in Table 2. The ways combined microbial metagenomes were constructed and utilized in later analyses are outlined in Table 3. Santa Pola and Isla Cristina metagenomes (SS13, SS19, IC21, SS33, and SS37) were analyzed individually except for assembly and *de novo* CRISPR detection steps. The Cahuil (C34) metagenome was analyzed individually throughout. All analyses performed on the microbial metagenomes are outlined in Fig. S1.

Haloviral genomes as well the Lake Tyrell viral contigs were downloaded from NCBI GenBank (Table S1). Additionally, Chula Vista saltern viral metagenomes were obtained through iMicrobe (*Hurwitz, 2014*) and then combined into one viral metagenome (Table 3). This combined viral metagenome (Table 3) was then assembled with Newbler version 2.9 into Chula Vista viral contigs that were also included in the haloviral library (Table S1).

## Profiling taxonomic and functional compositions of selected metagenomes

The selected individual metagenomes (Tables 1 and 2) were profiled for relative taxonomic abundances with MetaPhyler (*Liu et al., 2011a*). MetaPhyler aligns reads against a library of clade-specific marker genes. Reads that align against a marker gene above a particular threshold alignment score are then binned to the corresponding taxon (e.g., *Haloquadratum*, for reads aligning to *Haloquadratum* marker genes). Taxonomic profiles

**Table 3 Combined metagenomes, the individual metagenomes that compose them, and how they were used in following analyses.** The Cahuil metagenome (C34) was never included in any combined metagenome used for further analyses. The combined metagenomes were mainly used to improve *de novo* CRISPR detection, virus-host mapping using these *de novo*-detected CRISPR spacers, and assembly into contigs for detection of *cas* operons. These analyses required higher depth metagenomes; for example, no *de novo*-detected CRISPR spacers in several individual Santa Pola metagenomes mapped to any viruses in the halovirus library. The combined Chula Vista viral metagenome was used to assemble viral contigs included in the haloviral library (Table S1).

| Combined metagenome | Individual metagenomes included | How they were analyzed |
|---|---|---|
| Santa Pola and Isla Cristina (Combined SP) | SS13, SS19, IC21, SS33, and SS37 | • *De novo* CRISPR detection with Crass (**Fig. 4**)<br>• Reference-guided CRISPR detection (**Fig. 4**)<br>• Comparing DRs detected with the *de novo* CRISPR detection method Crass (**Fig. 6**)<br>• Virus-host mapping using *de novo*-detected CRISPR spacers (**Fig. 7**)<br>• Assembly to detect *cas* operons (**Fig. 10**) |
| Chula Vista (Combined CV) | CV6-8, CV12-14, and CV27-30 | • *De novo* CRISPR detection with Crass (**Fig. 4**)<br>• Reference-guided CRISPR detection (**Fig. 4**)<br>• Comparing DRs detected with the *de novo* CRISPR detection method Crass (**Fig. 6**)<br>• Assembly to detect *cas* operons (**Fig. 10**) |
| Chula Vista combined viral metagenome | Chula Vista viral metagenomes | • Assembly of Chula Vista viral contigs included in haloviral library (**Figs. 7** and **8**; **Table S1**) |

were further compared in MATLAB using principal component analysis (PCA). Clusters identified with PCA were evaluated for statistical significance using the ADONIS R function from the R package vegan (*Oksanen et al., 2016*) loaded in R version 3.2.2 (*R Core Team, 2015*). The significance of salinity in structuring the similarities between sample taxonomic profiles was determined using the envfit and ordisurf functions from the R package vegan (*Oksanen et al., 2016*), which fit vectors and smooth surfaces onto an ordination, respectively. Envfit was used with default parameters, while the number of knots (degrees of freedom plus one) for ordisurf was set to nine due to the nine metagenomes examined. The percent of reads mapping to markers for each metagenome were also calculated and compared to detect differences in the proportions of reads used for taxonomic profiling depending on metagenome.

The metagenomes were further profiled for functional abundances with ShotMAP (*Nayfach et al., 2015*) and metAnnotate (*Petrenko et al., 2015*). ShotMAP determination of gene family abundance was conducted with a Hidden Markov Model (HMM)-based search algorithm against an indexed library of Pfam family HMMs. Abundances were calculated from the numbers of metagenomic reads mapping to particular Pfam families normalized by the length of each Pfam target gene in the gene family. We selected abundances of nine gene families known to be related to glycerol metabolism for further analysis—dihydroxyacetone kinase (Dak1—PF02733; Dak1_2—PF13684; Dak2—PF02734), $NAD^+$-dependent glycerol dehydrogenase (Fe-ADH—PF00465; Fe-ADH_2—PF13685), $NAD^+$-dependent glycerol-3-phosphate dehydrogenase (both N- and C-terminal domains: NAD_Gly3p_dh_N–PF01210; NAD_Gly3p_dh_C–PF07479), and

glycerol kinase (FGGY_N–PF00370; FGGY_C–PF02782), listed both with Pfam family names and ID numbers. Relative abundances were then analyzed for correlations with respect to salinity using Microsoft Excel.

metAnnotate (*Petrenko et al., 2015*) was used to determine the abundance of particular Pfam gene families for particular taxa. Like ShotMAP, metAnnotate uses an HMM-based search algorithm to find reads that match certain Pfam gene family HMMs. Unlike ShotMAP, however, it also determines the taxonomic affiliations of these gene family reads, and thus determines which taxa have which gene families. We used metAnnotate to taxonomically profile nine aforementioned gene families known to be related to glycerol metabolism. We then calculated the proportion of reads mapping to each taxon among all those mapping to a gene family to determine the taxonomic profiles of these gene families. These gene family profiles were then visualized as heatmaps and hierarchically clustered with the R pheatmap package (*Kolde, 2015*). Clusters were evaluated for statistical significance with the ADONIS R function (*Oksanen et al., 2016*).

### *De novo* and reference-guided CRISPR detection

CRISPR arrays were detected using two different methods (reference-guided and *de novo*). All CRISPR detection analyses were treated as qualitative comparisons amongst sites because of limited sample size and inability to fully correct for differences in sequencing depth amongst metagenomes. Because CRISPR arrays' repetitive nature makes them highly difficult to assemble, CRISPR arrays were detected purely from unassembled metagenomic reads. *De novo* detection was performed with Crass (*Skennerton, Imelfort & Tyson, 2013*) with default parameters, which identifies repeat-spacer-repeat patterns in individual reads, uses these as "seeds" to recruit other CRISPR-containing reads, and then assembles these reads into CRISPR arrays. We further clustered Crass spacers using CD-HIT (*Li & Godzik, 2006*) with a similarity threshold of 0.9 and aligned them against known direct repeats from CRISPRdb (*Grissa, Vergnaud & Pourcel, 2007*) to determine the associated host organism.

Because Crass does not directly indicate the putative host encoding the detected CRISPRs, we developed a pipeline (MetaCRAST) to constrain spacer detection by expected host species (*Moller & Liang, 2016*). MetaCRAST searched each read for direct repeat (DR) sequences matching query DRs specified by the user. These query DRs were selected from known CRISPR DRs. In general, the query DR sequences might be known to be present in CRISPR arrays found in genomes expected from the taxonomic profile; for example, known genomic *Haloquadratum walsbyi* DRs might be used to search a metagenome for *Haloquadratum walsbyi* CRISPRs. Direct repeats within a certain Levenshtein edit distance (i.e., number of insertions, deletions, or substitutions necessary to convert one sequence to another) were identified in reads using the Wu-Manber multi-pattern search algorithm (*Wu, Manber & Myers, 1995*).

In the first step of the pipeline, reads that contain a single DR sequence matching the user-specified DR were detected with the Wu-Manber algorithm and collected for further analysis. In the second step of the pipeline, individual reads gathered in the first step were searched for two or more copies of the query direct repeat sequence within appropriate distances from each other. In the case of CRISPRs, direct repeats have been found to be

between 23 and 47 bp long, while spacers, and thus the distances between DRs, have been found to be between 26 and 50 bp long (*Skennerton, Imelfort & Tyson, 2013*). In the third step, the sequence fragments between the direct repeats were extracted as the spacers. These spacers were then collected as a non-redundant set in a hash, and then further clustered into non-redundant sets using CD-HIT (*Li & Godzik, 2006*).

MetaCRAST was implemented in Perl and takes a set of query direct repeats and a metagenome (each in FASTA format) as file inputs. It is publically available for download at GitHub (https://github.com/molleraj/MetaCRAST), where all command line parameters are also documented. It has also been validated against simulated metagenomes and tested against two real metagenomes corresponding to acid mine drainage (AMD) and enhanced biological phosphorus removal (EBPR) ecosystems (*Moller & Liang, 2016*).

We performed reference-guided CRISPR detection with MetaCRAST using the following parameters. We used a group of 29 DR sequences previously detected in Haloferacales and Halobacteriales genomes (Table S4) in the genomic CRISPR database CRISPRdb (*Grissa, Vergnaud & Pourcel, 2007*) as query DRs. We thus searched for reads containing sequences matching any of these 29 direct repeat sequences. The maximum allowed distance between detected direct repeats (i.e., maximum spacer length) was set to 60 bp. Acceptable edit distance—the allowed differences (insertions, deletions, or substitutions) between query and detected read DR sequences—were adjusted from 0 to 3 to see how variation in edit distance affected reference-guided CRISPR detection. All spacers detected for each condition (for each metagenome and acceptable error number) were then clustered using CD-HIT with a clustering similarity threshold of 0.9, which effectively clusters spacers together into a non-redundant set based on a pairwise similarity of 90% or greater.

## Mapping virus-host interactions with CRISPR spacers

CRISPR spacers extracted from the metagenomes with Crass and reference-guided methods and clustered with CD-HIT were then aligned against the library of haloviral genomes using NCBI BLAST (*Altschul et al., 1990*) with an e-value cutoff of 1e-03. All spacers detected with Crass for the combined Santa Pola dataset (see Table 3) were aligned against this library to create its corresponding interaction network. For several individual metagenomes (e.g., SS13, SS19), no Crass spacers mapped to the haloviral genomes, making the more sensitive reference-guided search (MetaCRAST) necessary. Thus, for individual metagenomes, spacers detected with MetaCRAST using a maximum Levenshtein edit distance of 3 were aligned for network creation. BLAST results from each method were then processed to form lists of virus-host pairs. Networks of virus-host interactions were drawn with Cytoscape (*Shannon et al., 2003*).

## Comparative analysis of virus and host sequence characteristics

Viruses and hosts matched through CRISPR mapping were further compared in nucleotide k-mer frequencies to determine whether virus and host frequencies were similar. Dinucleotide and trinucleotide usages were determined using the function *nmercount* from the MATLAB Bioinformatics Toolbox (*The MathWorks, Inc., 2015*). Dinucleotide and trinucleotide frequencies were compared for a non-redundant list of viruses associated

to each of the six detected hosts listed below. All viruses were associated to possible hosts using our in-house reference-based CRISPR detection method. *Haloquadratum walsbyi* C23, *Haloferax volcanii* DS2, *Halogeometricum borinquense* DSM 11551, *Halorhabdus utahensis* DSM 12940, *Haloarcula marismortui* ATCC 43049 (chromosome 1), and *Haloarcula hispanica* N601 (chromosome 1) host genomes were analyzed along with the matched haloviral genomes. To measure the sequence characteristic differences among virus-host pairs, each virus dinucleotide and trinucleotide frequency was subtracted from each corresponding host genome frequency and squared. All these squared distances were summed for possible dinucleotides and for possible trinucleotides to generate an overall Euclidean measure of distance between virus and host.

## Assembling metagenomes and detecting halobacterial *cas* genes in metagenomic contigs

Combined Santa Pola, combined Chula Vista, and Cahuil metagenomes (Table 3 outlines combined metagenomes) were assembled into contigs to detect *cas* operons. Combined Santa Pola, combined Chula Vista, and Cahuil metagenomes were assembled both with Newbler and Velvet, creating a total of six sets of contigs (*Zerbino & Birney, 2008*). Both Newbler and Velvet were used to compare the effects of Overlap-Layout-Consensus (Newbler) and de Bruijn graph (Velvet) assembly methods. Newbler assembly was performed with default parameters, while Velvet assembly was performed with a k-mer size of 31.

Newbler and Velvet contigs assembled from each combined metagenome were first aligned against *cas* genes annotated in *Haloferax volcanii* DS2, *Halorubrum lacusprofundi* ATCC 49239, and *Haloquadratum walsbyi* C23 genomes using NCBI blastn (*Altschul et al., 1990*) with an e-value cutoff of 1e-06 and NCBI tblastx with an e-value cutoff of 1e-30. Color-coded *cas* detection results for genes in the *Haloquadratum*, *Halorubrum*, and *Haloferax cas* operons were drawn with the synthetic biology tool Pigeon (*Bhatia & Densmore, 2013*). A library of *cas* genes from the three major halobacterial orders (the Haloferacales, Halobacteriales, and Natrialbales) were downloaded from NCBI Genbank (see Tables S11–S13) and aligned against each set of assembled contigs (assembled with each method and from each combined metagenome) to determine the taxonomic affiliation of matching *cas* genes. BLAST alignments were performed with the same parameters described above. Taxonomy information was extracted from matching *cas* gene Genbank records with the NCBI Efetch utility to determine the taxonomic affiliations of the assembled *cas* contigs.

## RESULTS AND DISCUSSION

### Taxonomic profiling identifies predominant members of saltern microbial communities

We profiled the taxonomic abundances of metagenomes sequenced from **C**ahuil (C34), **C**hula **V**ista (CV6-8, CV12-14 and CV27-30), **I**sla **C**ristina (IC21), and **S**anta Pola **S**altern (SS13, SS19, SS33, SS37) sites (where the ID numbers indicate salinity percentages) to better understand the population structures of saltern microbial communities and to

compare host abundances amongst sites. We found the low salinity Chula Vista (CV6-8) and SS13 salterns had different microbial communities from the other salterns, containing large proportions of bacterial Actinobacteria, Proteobacteria and Rhodobacteraceae (7.64, 14.28, and 20.69% in SS13 and 6.6, 10.05, and 11.13% in CV6-8, respectively—see Fig. 1A). In addition, we found *Haloquadratum* was the major taxon identified in the SS19, SS33, and SS37 salterns (88.32, 65.55, and 78.06% abundance, respectively), while *Halorubrum* and Halobacteriaceae were major components of the Cahuil and IC21 salterns (31.79 and 64.55% in Cahuil; 32.5 and 51.63% in IC21, respectively–see Fig. 1A). Bacterial *Salinibacter* was identified in the SS19, SS33, and SS37 salterns (2.18, 7.16, and 6.82% relative abundances, respectively). While *Haloquadratum* was the major identified taxon in the higher salinity Santa Pola salterns, Halobacteriaceae composed the most abundant identified taxon identified in the medium and high salinity Chula Vista salterns (51.46 and 77.02%, respectively). We realize that the Halobacteriaceae component of the saltern communities may also include species in the new order Haloferacales (*Gupta, Naushad & Baker, 2015*), but this taxon was defined after the MetaPhyler marker library was constructed (*Liu et al., 2011a*). These taxonomic profiles and clustering results further guided our analyses of host CRISPR arrays, as we used taxonomy to guide our searches for CRISPR spacers.

We further compared the taxonomic profiles through principal component analysis, which found that the first two principal components accounted for 88% of variation amongst the examined taxonomic profiles. PCA grouped the SS19, SS33, and SS37 profiles; low salinity Chula Vista (CV6-8) and SS13 profiles; and medium salinity Chula Vista (CV12-14), high salinity Chula Vista (CV27-30), IC21, and C34 profiles (Fig. 1B). These groups were found to be statistically significant in their between-group differences (ADONIS; F=19.817; $p = 0.002$). We furthermore found percent salinity to be a significant feature in structuring the similarities amongst metagenomes when fitting this variable to the PCA ordination either as a vector or a response surface (envfit: $p = 0.038$; ordisurf, knots=9: $p = 0.0291$).

We also determined the percentage of reads mapping to phylogenetic markers to detect read length-dependent differences in taxonomic profiling among metagenomes (Fig. S2). The percentage mapping was similar among Santa Pola, Isla Cristina, and Cahuil (C34) metagenomes (an average of 0.51% of reads with a standard deviation of 0.0483% among these metagenomes), but lower among the Chula Vista metagenomes (an average of 0.21% of reads with a standard deviation of 0.041% among these metagenomes) due to lower coverage (see Tables 1 and 2).

We have thus built upon prior metagenomics studies to show that *Haloquadratum* is the majority taxon in many salterns (*Oh et al., 2009*), particularly those that are environmentally stable with high salinity (Fig. 1). The Santa Pola and Isla Cristina metagenomes have previously been taxonomically profiled through mapping of reads to 16S rRNAs obtained from the Ribosomal Database Project (*Fernandez et al., 2013*; *Fernández et al., 2014b*; *Ventosa et al., 2014*). These 16S rRNA profiling studies revealed that the lowest salinity saltern (SS13) contained a community of bacterial Actinobacteria and Proteobacteria and archaeal Euryarchaeota, while members of the Euryarchaeota

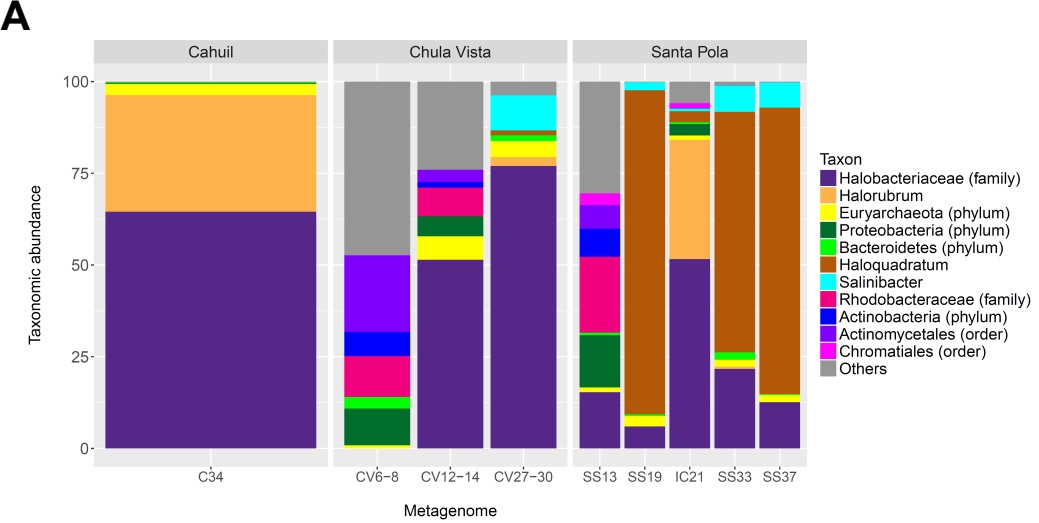

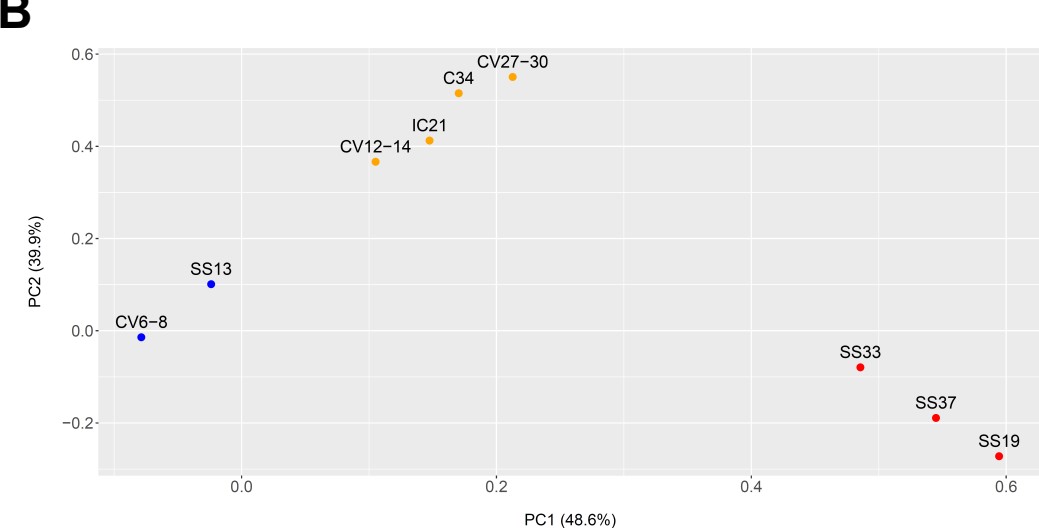

**Figure 1 Taxonomic profiles (A) and principal component analysis (B) of selected saltern metagenomes.** Metagenomes were profiled using MetaPhyler and resulting profiles clustered using MATLAB. Profiles show the relative abundances (in percent) of 11 different taxa identified with MetaPhyler (A). Profiles are grouped by geographic locations—Cahuil, Chula Vista, and Santa Pola (A). The first two principal components, which together account for 88% of variation in the profiles, are shown in the PCA results (B). Three different groups of similar profiles were identified with PCA (blue—CV6-8, SS13; orange–CV12-14, IC21, C34, CV27-30; red—SS33, SS37, SS19). These groups were found to be statistically significant in their between-group differences (ADONIS; $F = 19.817$; $p = 0.002$).

increased in abundance with increasing salinity (*Fernández et al., 2014c*; *Fernández et al., 2014a*; *Ventosa et al., 2015*). Compared with the Santa Pola salterns, Cahuil and Isla Cristina salterns may be more dynamic, with more rapidly changing daily and yearly temperatures and higher rainfall (*Fernández et al., 2014c*). This may lead to lower concentrations of potassium, magnesium, and sulfate in those salterns, thus leading to lower abundances of *Haloquadratum*, and higher abundances of other haloarchaea (*Podell et al., 2014*; *Fernández et al., 2014c*).

## Functional profiling reveals abundances of glycerol metabolism genes and their taxonomic affiliations

Glycerol is the main carbon source for saltern microbes (*Oren, 1993*). We used two functional profiling tools to study the glycerol metabolism genes critical for carbon cycling in these ecosystems. We used the tool ShotMAP to determine abundances of glycerol metabolism gene families (*Nayfach et al., 2015*) and metAnnotate to study the taxonomic distribution of reads mapping to these gene families (*Petrenko et al., 2015*). As with MetaPhyler, Cahuil, Isla Cristina, and Santa Pola metagenomic reads mapped to the ShotMAP reference library in higher proportions than the Chula Vista reads (Fig. S3). Using these tools, we examined the abundances and taxonomic affiliations of the glycerol kinase (FGGY_N and FGGY_C), NAD$^+$-dependent glycerol-3-phosphate dehydrogenase (NAD_Gly3P_dh_N and NAD_Gly3P_dh_C), glycerol dehydrogenase (iron-containing alcohol dehydrogenase –Fe-ADH and Fe-ADH_2), and dihydroxyacetone kinase (Dak1, Dak1_2, and Dak2) Pfam gene families.

We first determined gene family abundances with ShotMAP and then examined relationships between these abundances and salinity levels. Among the gene families we examined, we noticed both negative and positive correlations between relative gene family abundance and salinity (Fig. 2). Dak1, Dak2, FGGY_N, and Fe-ADH_2 families showed positive correlations with salinity, while Dak1_2, NAD_Gly3P_dh_N, NAD_Gly3P_dh_C, FGGY_C, and Fe-ADH families showed negative correlations with salinity. Only Dak1_2, NAD_Gly3P_dh_N, and NAD_Gly3P_dh_C gene families showed significant correlations with salinity ($r = -0.617, -0.795$, and $-0.787$; $p = 0.038, 0.005$, and $0.006$, respectively). We also noted family abundances calculated for the Chula Vista metagenomes to be larger than those found for any other metagenome. More short reads mapping to these gene families may account for these higher family abundances than those found for the Santa Pola, Isla Cristina, and Cahuil metagenomes.

We next determined the taxonomic affiliations of reads mapping to these gene families using metAnnotate. We generated heat maps of these affiliations hierarchically clustered both with regard to metagenome (Fig. 3A) and gene family (Fig. 3B). As with taxonomic profiles, the taxonomic affiliations of glycerol metabolism genes in the *Haloquadratum*-rich SS19, SS33, and SS37 metagenomes group together, as do those of the Chula Vista metagenomes, and the SS13, IC21, and Cahuil (C34) metagenomes (Fig. 3A). Across gene family, taxonomic affiliations of the dihydroxyacetone kinase families were closest to each other, as well as those of the alcohol dehydrogenase (Fe-ADH) and glycerol kinase (FGGY) gene families (Fig. 3B). The two NAD$^+$-dependent glycerol-3-phosphate dehydrogenase families grouped together, but many reads mapping to these families were found to be unclassified (Fig. 3B). These described gene family and metagenome clusters were found to be statistically significant (ADONIS; $F = 7.5468$ and $2.8768$; $p = 0.014$ and $0.013$, for gene family and metagenome clusters, respectively).

We also analyzed these taxonomic affiliations for each gene family across each individual metagenome (Fig. 3C). *Haloquadratum* associated with dihydroxyacetone kinase gene families in the SS19, SS33, and SS37 metagenomes while *Halorubrum* associated with these families in the IC21 and Cahuil metagenomes (Fig. 3C). The two NAD$^+$-dependent

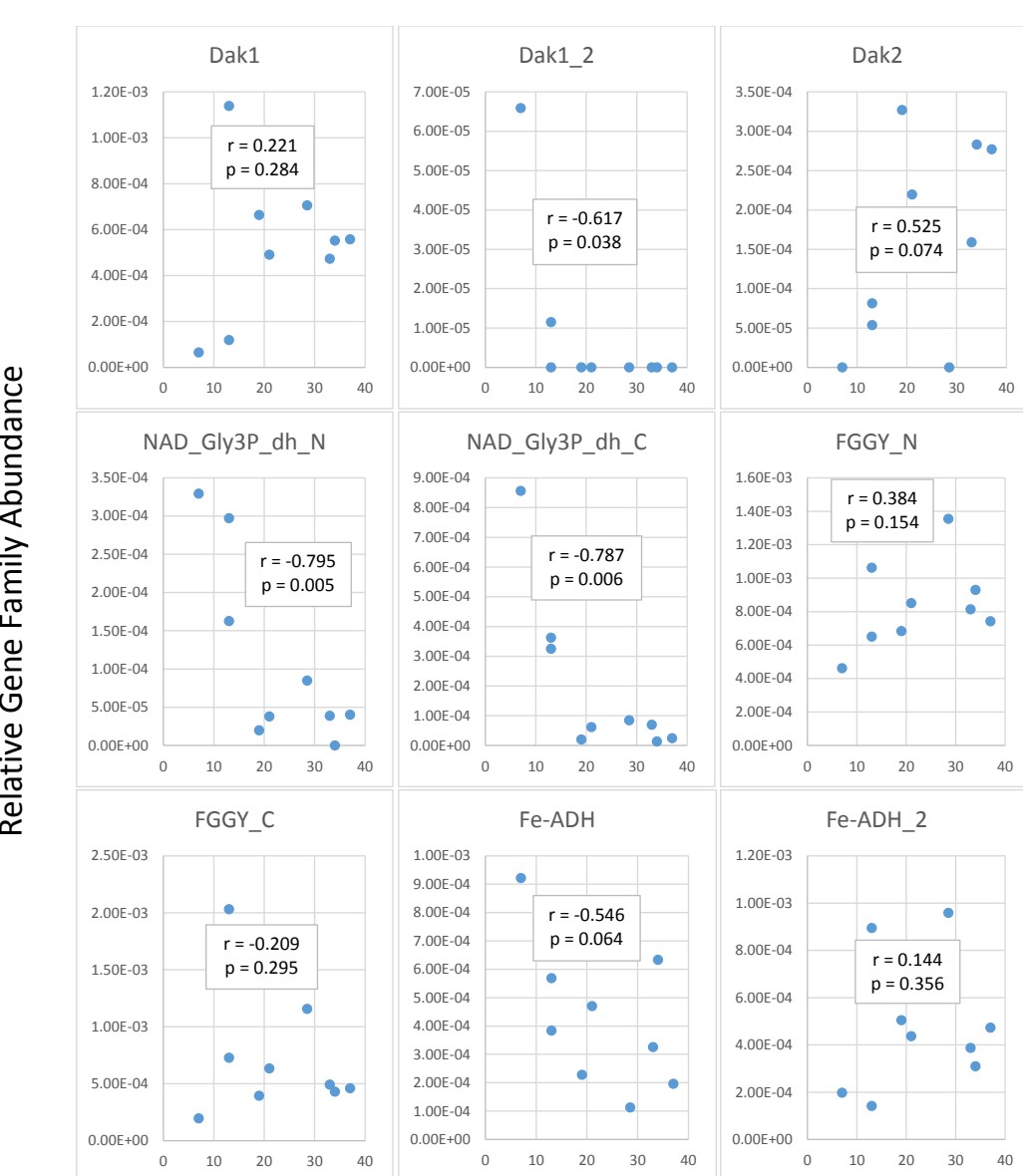

**Figure 2 Analysis of correlations between percent salinity and relative gene family abundances determined with ShotMAP.** We determined relative abundances of nine different gene families using ShotMAP (glycerol kinase, FGGY_N and FGGY_C; NAD$^+$-dependent glycerol-3-phosphate dehydrogenase, NAD_Gly3P_dh_N and NAD_Gly3P_dh_C; glycerol dehydrogenase/iron-containing alcohol dehydrogenase, Fe-ADH and Fe-ADH_2; and dihydroxyacetone kinase, Dak1, Dak1_2, and Dak2). Relative gene family abundance (listed on the $y$-axis) was calculated as the sum of hits to target genes within a gene family each normalized by target gene length. Correlations between relative gene family abundance and percent salinity (weight per volume) and respective $p$-values were determined with Microsoft Excel.
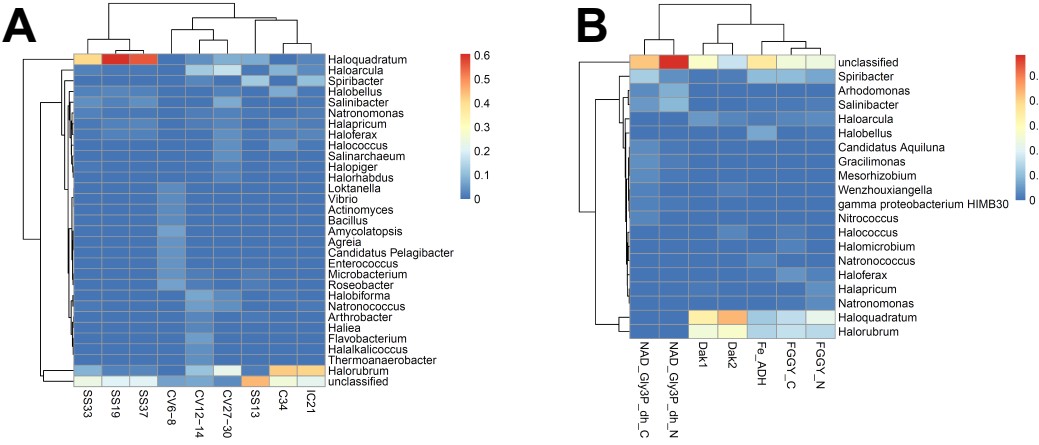

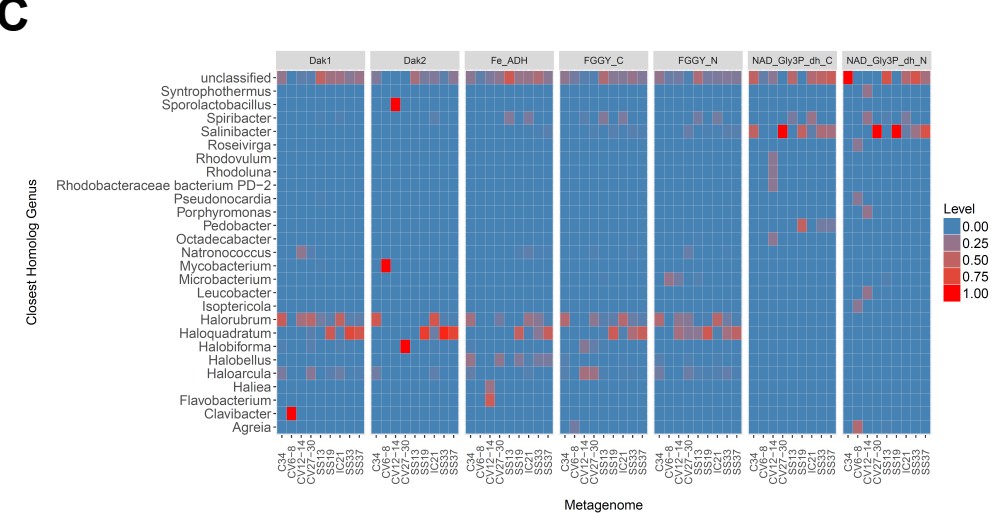

**Figure 3** **Heat map visualizations of glycerol metabolism taxonomic profiles determined with metAnnotate.** We determined taxonomic profiles of nine different gene families using metAnnotate (glycerol kinase, FGGY_N and FGGY_C; NAD$^+$-dependent glycerol-3-phosphate dehydrogenase, NAD_Gly3P_dh_N and NAD_Gly3P_dh_C; glycerol dehydrogenase/iron-containing alcohol dehydrogenase, Fe-ADH and Fe-ADH_2; and dihydroxyacetone kinase, Dak1, Dak1_2, and Dak2). Relative gene family abundance (referred to as level in C) here is defined as the fraction of all hits for a gene family mapping to a particular taxon. Abundance profiles were determined for genus and site (A), genus and gene family (B), and genus, site, and gene family (C). Only site and gene family profiles (A and B) were hierarchically clustered. The primary gene family and metagenome (site) clusters (see Results) were found to be statistically significant (ADONIS; $F = 7.5468$ and $2.8768$; $p = 0.014$ and $0.013$, for gene family and metagenome clusters, respectively). The first two heat maps (A and B) were generated with the heatmap R package, while the last heat map (C) was generated with ggplot2.

glycerol-3-phosphate dehydrogenase families, on the other hand, associated mainly to *Salinibacter* in the Cahuil (C34), high salinity Chula Vista, SS19, SS33, and SS37 metagenomes and to *Spiribacter* in the SS13 and IC21 metagenomes. Unlike the previous families, alcohol dehydrogenase and glycerol kinase had less clear associations across the metagenomes, with association to *Haloquadratum* and *Halorubrum* but also many

unclassified reads. In addition, glycerol kinase families associated to *Spiribacter* (SS13 and IC21) and the alcohol dehydrogenase family to *Halobellus* (C34, CV27-30, SS19, SS33, and SS37) across several metagenomes.

Our functional studies thus found both positive and negative correlations between glycerol metabolism gene family abundances and salinity (Fig. 2). Prior studies of the Santa Pola and Isla Cristina salterns found that glycerol metabolism genes, such as dihydroxyacetone kinase, glycerol kinase, and NAD$^+$-dependent glycerol-3-phosphate dehydrogenase, increased in abundance with salt concentrations (*Fernández et al., 2014c*; *Fernández et al., 2014a*). Taxonomic affiliations, however, differed amongst salterns and different gene families (Fig. 3). Dihydroxyacetone kinase most strongly associated with *Haloquadratum*, except in the IC21 and Cahuil (C34) salterns, where it associated strongly with *Halorubrum* instead. NAD$^+$-dependent glycerol-3-phosphate dehydrogenase, on the other hand, associated with *Salinibacter* (Fig. 3C). Based on prior understanding of saltern glycerol cycling (*Oren, 1993*; *Oren, 1994*; *Sher et al., 2004*), these results together suggest archaea such as *Haloquadratum* and *Halorubrum* consume the glycerol metabolism intermediate dihydroxyacetone in these ecosystems, while bacteria such as *Salinibacter* may first phosphorylate and then oxidize glycerol to metabolize it (*Sher et al., 2004*).

## Metagenomic CRISPRs detected with two independent methods show variation in CRISPR arrays depending on community composition

We detected CRISPR arrays in metagenomes using both *de novo* and reference-guided methods. We used Crass for *de novo* detection (*Skennerton, Imelfort & Tyson, 2013*) and a custom pipeline (MetaCRAST) for reference-guided detection (see Materials and Methods). We used a combination of Halobacteriales and Haloferacales CRISPR direct repeats downloaded from CRISPRdb as our query for MetaCRAST analyses across all metagenomes.

Overall, more spacers were detected in IC21 and Cahuil (C34) metagenomes compared with the high salinity Chula Vista, SS13, SS19, SS33, and SS37 metagenomes (Fig. 4) both with MetaCRAST (3 errors allowed) and Crass. More spacers were detected with Crass compared to reference-guided methods in the combined Santa Pola, SS13, and SS33 metagenomes (377 vs. 277, 29 vs. 9, and 146 vs. 80 spacers, respectively). For the reference-guided detection analyses, the number of spacers identified increased as the allowable error (insertions, deletions, or substitutions different from the query DR sequence) in alignment was increased from 0 to 3 (Fig. 4). No spacers were detected with either method in the low or medium salinity Chula Vista metagenomes. When we normalized by total number of metagenomic reads, we found that the Cahuil (C34) metagenome had the largest number of spacers detected per million reads (Fig. S4). Nonetheless, it is important to note that CRISPR detection analyses could only be treated as qualitative rather than quantitative because of limited sample size and inability to fully correct for differences in sequencing depth amongst metagenomes.

We then compared the detected direct repeats and spacers among metagenomes to discover common CRISPR array types or spacers (Figs. 5 and 6). Remarkably, common spacers were identified amongst the SS19 and SS37 metagenomes (13 with Crass and 18

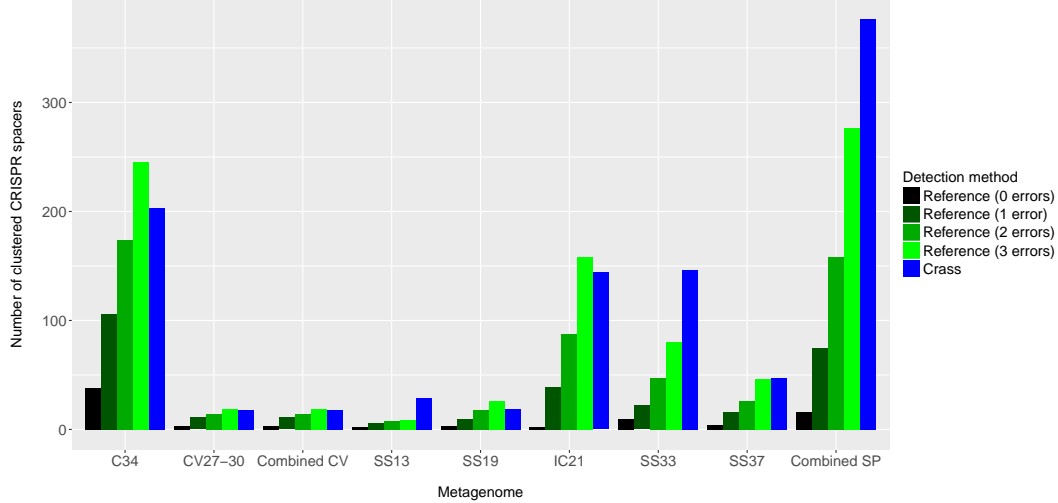

**Figure 4  Number of CRISPR spacers detected in each metagenome after clustering with CD-HIT.** Spacers were detected either with the *de novo* detection method (Crass) or the reference-guided method (MetaCRAST) with corresponding maximum edit distances (from 0 to 3) described in 'Materials and Methods' that used a query of 29 halobacterial CRISPR direct repeat sequences. All spacer counts are reported after clustering initially detected spacers with CD-HIT (with a clustering similarity threshold of 0.9).

with MetaCRAST among these two, respectively) using both detection methods (Figs. 5A and 5B). Amongst the metagenomes with a higher number of detected spacers (IC21 and Cahuil/C34), common spacers were detected with both methods (5 in IC21 and 24 in Cahuil—see Figs. 5C and 5D). Amongst the direct repeats detected with Crass in the combined Chula Vista, combined Santa Pola, and Cahuil metagenomes, there was one direct repeat (DR) sequence common to both Cahuil and combined Santa Pola (Fig. 6). We also searched the combined Santa Pola direct repeats against a full database of direct repeats from CRISPRdb to determine possible taxonomic affiliations. We found that 14 of the 67 detected combined Santa Pola DRs matched DRs in the CRISPRdb database (Table S3). Of these 14 matching DRs, 10 matched haloarchaeal DRs, while 4 matched bacterial DRs.

Our CRISPR detection results showed a smaller number of CRISPR spacers found in salterns where *Haloquadratum* was identified as the majority taxon compared with salterns where Halobacteriaceae was found to be the majority taxon. Furthermore, comparison of detected metagenomic spacers to those found in the genomes of two different *H. walsbyi* strains (C23 and DSM 16790) suggests variation in CRISPR array length in the salterns. Strain C23 contains a full set of *cas* genes and several CRISPR arrays, while strain DSM 16790 lost all its *cas* genes and all but five spacers (*Dyall-Smith et al., 2011*). While we could not identify any C23 spacers amongst our metagenomes, we could find at least three DSM 16790 spacers in several metagenomes. A successful mobile element integration event in spite of CRISPR defense is thought to have promoted loss of spacers in *Haloquadratum* genomes (*Dyall-Smith et al., 2011*). Taken together, this evidence suggests genomic CRISPR arrays in *Haloquadratum* species may vary in length.

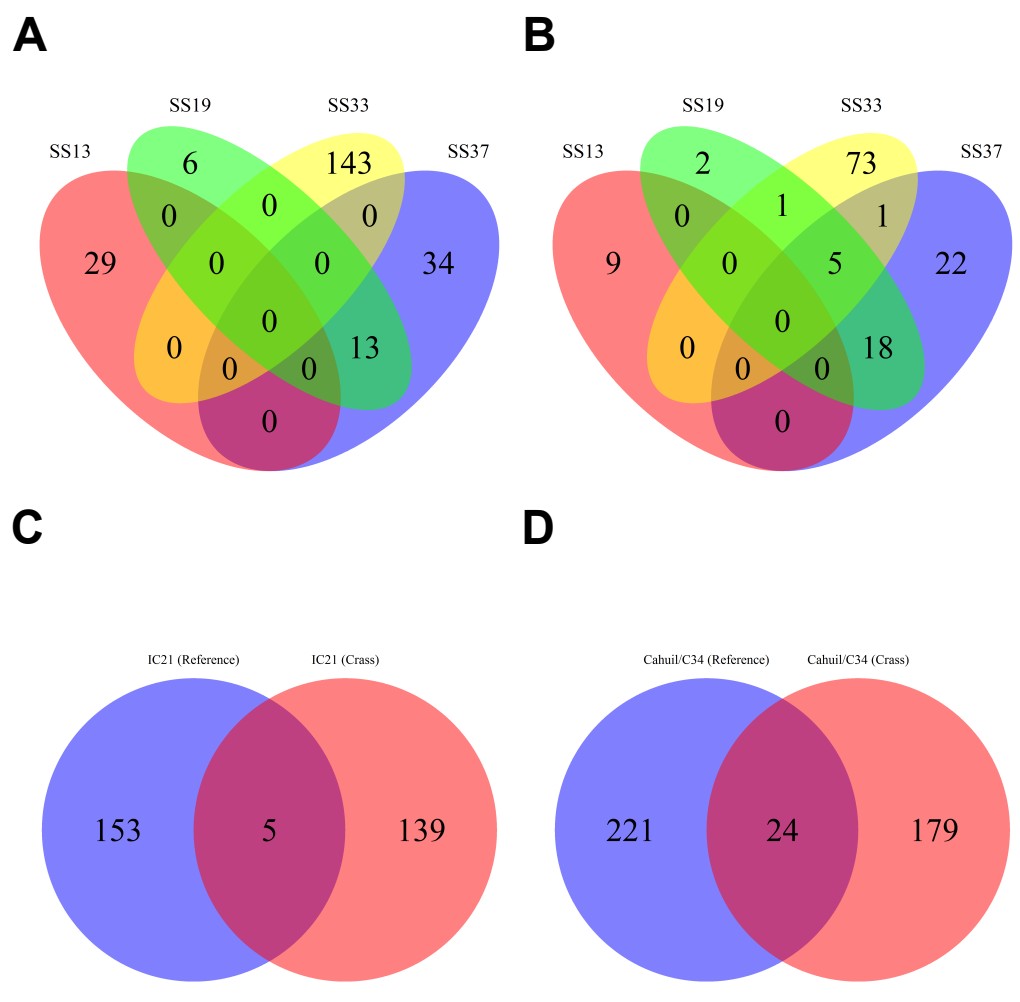

**Figure 5** **Comparison of CRISPR spacers detected with *de novo* (Crass) and reference-guided (MetaCRAST) methods.** All reference-guided spacers compared here with Crass spacers were detected with a maximum allowed edit distance of 3 (insertions, deletions, or mismatches with query direct repeat sequence). CRISPR spacers detected with *de novo* (A) and reference-guided (B) methods in SS13, SS19, SS33, and SS37 metagenomes were compared amongst sites to determine identical spacers. For the IC21 (C) and Cahuil/C34 (D) metagenomes, spacers were compared between the two detection methods.

## Metagenomic CRISPRs detected with two independent methods map haloviruses to saltern hosts

We then aligned the spacers detected with both methods against a library of haloviral genomes (Table S1) to determine virus-host interactions in these salterns. Unlike CRISPR spacers detected with Crass in several individual Santa Pola metagenomes, spacers detected in a combination of all Santa Pola and Isla Cristina metagenomes did align to haloviral genomes (Fig. 7). In this Crass analysis of the combined Santa Pola dataset, we detected virus interactions most likely with *Haloquadratum* (G467), *Haloarcula* (G42 and G82), and *Saccharomonospora* (G57) hosts, based on our previous comparison of Crass DRs (listed by group number, such as G467) against CRISPRdb DRs (Fig. 7). For many of the interactions (e.g., G467, G42), we could determine a possible host by such comparison. For several

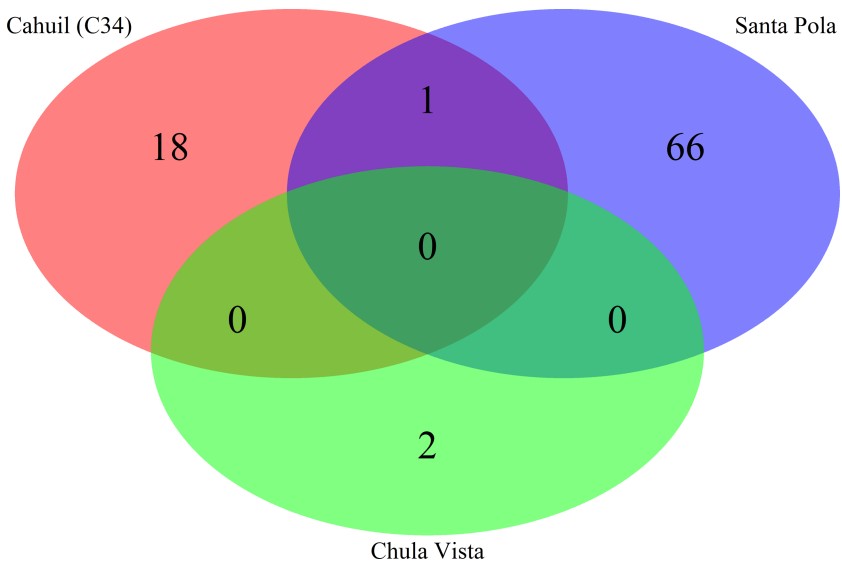

**Figure 6** **Comparison of CRISPR direct repeats detected with the** *de novo* **(Crass) method.** Direct repeats detected with Crass were compared amongst Cahuil (C34), combined Chula Vista, and combined Santa Pola and Isla Cristina metagenomes listed in Table 3.

notable groups of interactions, however, we could not determine possible hosts (e.g., G85, G1290).

While Crass only provided a picture of virus-host interactions in the combined Santa Pola metagenome, MetaCRAST detected spacers that mapped to haloviral genomes in each of the individual Santa Pola, Isla Cristina, and Cahuil metagenomes (Fig. 8). Only one interaction was detected in the SS13 dataset (Fig. 8A), but there were strikingly similar groups of virus-host interactions detected in the SS19, SS33, and SS37 metagenomes (Figs. 8B– 8D). While many groups of interactions were similar to those expected from the *Haloquadratum walsbyi* strain C23 CRISPR array (Fig. S5), no C23 spacers matched any detected metagenomic spacers, while at least three *Haloquadratum walsbyi* strain DSM 16790 spacers did match spacers detected in SS19, IC21, SS33, and SS37 metagenomes. Aside from *Haloquadratum*, in the SS33 metagenome, we also detected *Haloarcula*-virus interactions with eHP-4, eHP-8, and eHP-13 (Fig. 8C). Quite surprisingly, we noticed remarkably different virus-Halobacteriaceae interactions in the IC21 and Cahuil (C34) metagenomes (Figs. 8E and 8F), perhaps due to differences in taxonomic composition at the genus level.

Our CRISPR mapping results not only showed consistent viral interactions with *Haloquadratum* across salterns but also numerous interactions with less abundant taxa such as *Haloferax* and *Haloarcula*. As expected from our comparison of taxonomic and functional profiles, there were consistent sets of interactions amongst SS19, SS33, and SS37 salterns, where *Haloquadratum* species form the majority of the microbial population (Fig. 8). These interactions suggest similar viral infection patterns across these environments, consistent with "global pond" theories of microbial and viral exchange (*Dyall-Smith et al., 2011*; *Atanasova et al., 2012*). Unclear biological requirements for CRISPR spacer

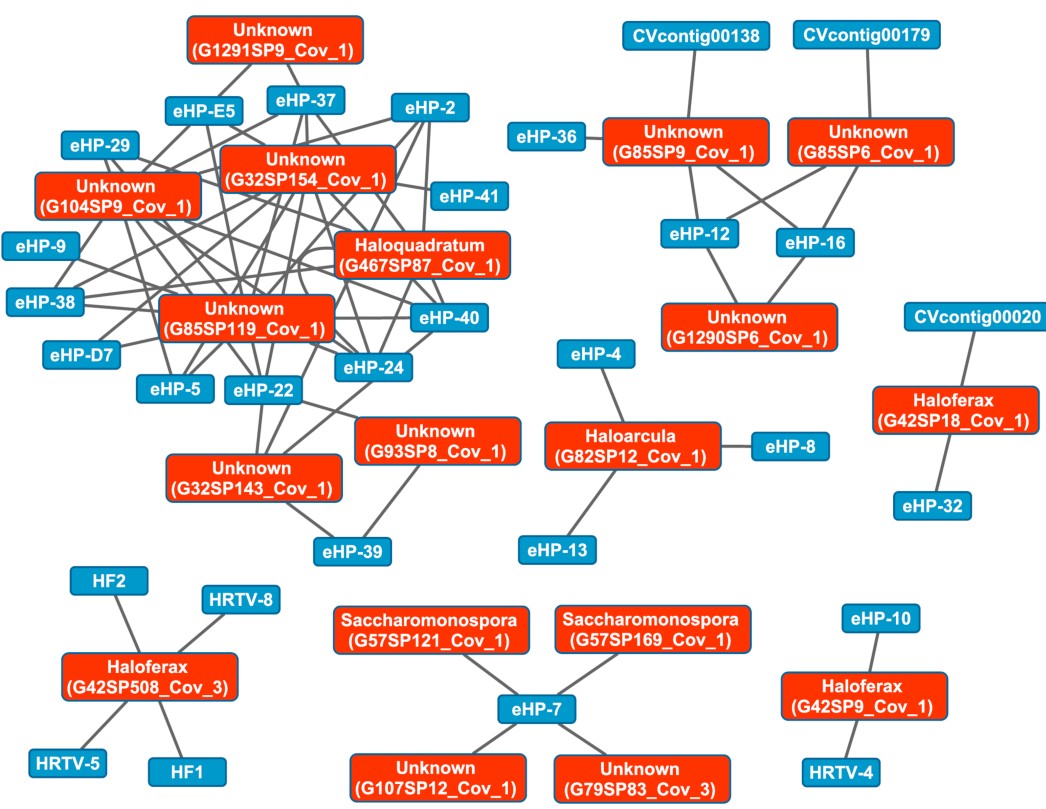

**Figure 7** **Map of virus-host interactions generated by aligning Crass spacers detected in the combined Santa Pola and Isla Cristina metagenome (Table 3) against a library of haloviral genomes.** The library of haloviral genomes screened is listed in Table S1. Nodes represent either viruses or spacers, while edges represent BLAST alignments linking spacers to viruses. Viruses are marked in blue and spacers in orange. Spacers are listed with both the most likely taxonomic affiliation and sequence ID number (in parentheses). Visualization was performed with Cytoscape.

incorporation make it difficult to make any conclusion about the abundances of the viruses. Though some models suggest viruses must have reached a critical number within their hosts to induce CRISPR spacer incorporation into host genomes (*Garrett, Vestergaard & Shah, 2011*), characterization of the *Haloferax volcanii* CRISPR systems suggests plasmid copy number has little effect on incorporation (*Maier et al., 2013*).

There were also interactions detectable with Crass that could not map to any known hosts (Fig. 7). Some of these hosts may be poorly characterized members of the Nanohaloarchaea, as recent studies have suggested for eHP-12 and eHP-16 (*Martínez-García et al., 2014*). Surprisingly, we did not detect any spacers or virus-host interactions with *Halorubrum* with either CRISPR detection method. Either limitations of our detection tools or a lack of *Halorubrum* CRISPRs could account for this observation.

## Nucleotide usage comparison suggests biological plausibility of virus-host associations

To determine whether our virus-host mappings are biologically plausible, we compared the dinucleotide and trinucleotide frequencies of CRISPR-determined virus-host pairings. Over

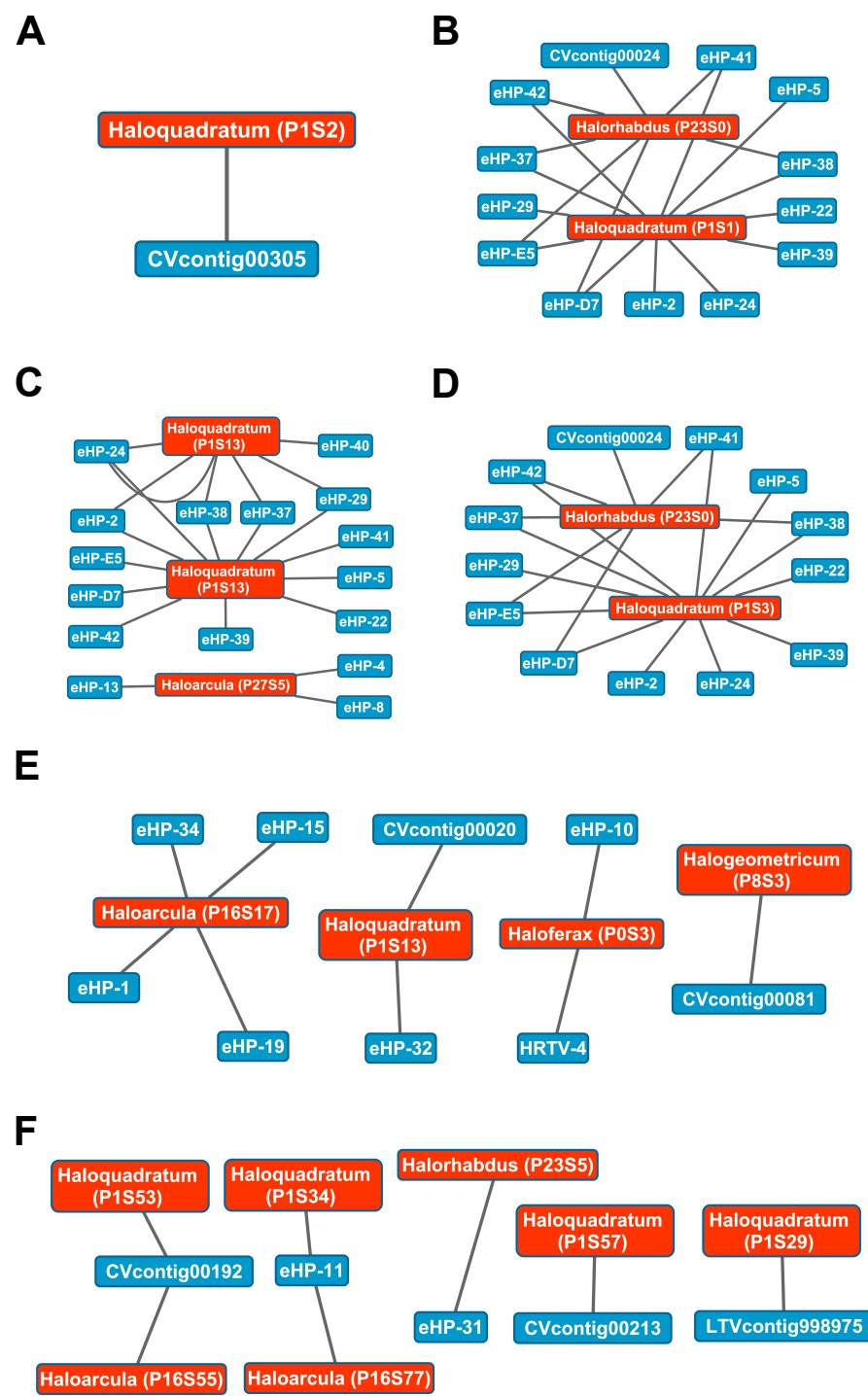

**Figure 8 Map of virus-host interactions generated by aligning spacers detected with reference-guided methods in SS13 (A), SS19 (B), SS33 (C), SS37 (D), IC21 (E), and Cahuil/C34 (F).** The library of haloviral genomes screened is listed in Table S1. Nodes represent either viruses or spacers, while edges represent BLAST alignments linking spacers to viruses. Viruses are marked in blue and spacers in orange. Spacers are listed with both the most likely taxonomic affiliation and sequence ID number (in parentheses). Visualization was performed with Cytoscape.

evolutionary time, viruses' nucleotide usage steadily approaches that of the host, reflecting adaptation to host codon usage, tRNA availability, intracellular nucleotide pools, and host restriction modification systems (*Edwards et al., 2015*). We compared these nucleotide usage frequencies both among the predicted hosts and analyzed metagenomes (Fig. 9). Among predicted hosts, *Haloquadratum walsbyi*, *Haloferax volcanii*, and *Haloarcula marismortui* showed the smallest overall differences in dimer and trimer usage, suggesting strong virus-host association, while *Haloarcula hispanica* and *Halorhabdus utahensis* showed the highest overall differences (Figs. 9A and 9B). Only one virus associated to *Halogeometricum borinquense*, and it had a greater difference in trimer frequency compared with dimer frequency relative to the other predicted hosts. Among the metagenomes, the smallest differences were found in the SS33 metagenome; the next smallest differences were found in the Cahuil (C34), SS19, and SS37 metagenomes, while the largest difference was found in the single virus-host pair detected in the SS13 metagenome (Figs. 9C and 9D).

## Assembled metagenomic contigs contain halobacterial *cas* genes

To further confirm our CRISPR detection and mapping results, we searched through metagenomic contigs for the *cas* genes required for CRISPR activity. Because all microbial metagenomes studied were sequenced with 454 technology, and Newbler is designed to assemble 454 reads, we expected longer contigs to be assembled with Newbler and found this in our assembly results (Table S2). We both searched for contigs that matched several different haloarchaeal *cas* operons (Fig. 10A) and profiled the taxonomic affiliations of haloarchaeal *cas* genes that best matched contigs (Fig. 10B). We were able to assemble almost all genes in *cas* operons in three different metagenomes (Cahuil, combined Chula Vista, and combined Santa Pola) from several different Type I-B and Type I-D *cas* operons (Fig. 10A).

We also searched a library of *cas* genes from three different haloarchaeal orders (Halobacteriales, Haloferacales, and Natrialbales) and generated profiles of taxonomic affiliations for the genes that best matched contigs (Fig. 10B). *Haloarcula* composed a large proportion of Halobacteriales contig affiliations, especially amongst the Cahuil (C34) contigs (71.4% among Newbler contigs; 41.4% among Velvet contigs). On the other hand, *Haloquadratum* and *Halorubrum* composed large proportions of the Haloferacales affiliations, especially for the combined Santa Pola dataset (14.5% and 25.8% for Newbler contigs; 23.5% and 26.6% for Velvet contigs). We also calculated the total number of hits in the search against the *cas* gene libraries (Fig. S6) and profiled the taxonomic affiliations of all *cas* gene alignments to the contigs (Fig. S7). There were smaller *Haloarcula*, *Halonotius*, *Halorhabdus*, and *Haloquadratum* proportions in the total hit profiles (Fig. S7) compared with the best hit profiles (Fig. 10B).

## CONCLUSIONS

Taken together, our results suggest common sets of virus-host interactions across different salterns and variation in CRISPR antiviral defenses amongst saltern microbes. While common sets of viruses infect haloarchaea across salterns, these hosts may have strikingly

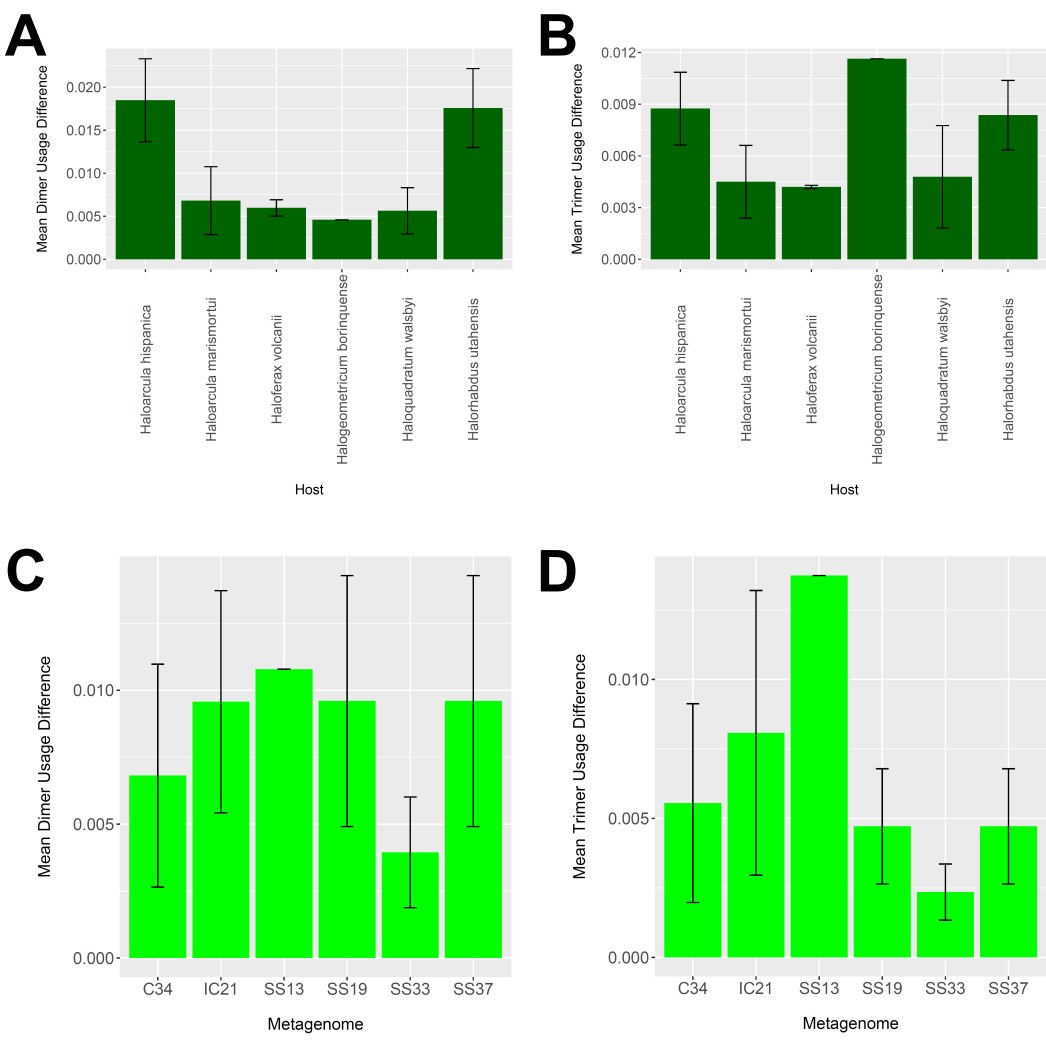

**Figure 9** **Differences in dimer and trimer usages between host and virus genomes matched with CRISPR spacers.** The difference between each possible dimer and trimer usage was subtracted, squared, and summed across all possible dimers or trimers to generate a measure of dimer or trimer difference. The average dimer usage and trimer usage differences were calculated for all combinations of viruses with a particular host (A and B, respectively) and all virus-host combinations detected for a particular metagenome (C and D, respectively). All averages are reported with a 95% confidence interval (within two standard error) in black.

different levels of CRISPR defenses. *Haloquadratum*, it seems, may exist in either virus-resistant or virus-sensitive strains depending on the presence or absence of its CRISPR/Cas system. CRISPR/Cas systems may be part of the previously examined *Haloquadratum* accessory gene pool (*Legault et al., 2006*). While it is surprising that virus-sensitive strains would be present in such a virus-rich environment, even these forms may have some innate halovirus resistance through extensive genomic variation of phage recognition proteins (*Cuadros-Orellana et al., 2007*). These patterns may not apply to other haloarchaea and saltern inhabitants, however. While *Haloquadratum* encodes CRISPRs chromosomally, *Haloarcula* and *Haloferax* encode CRISPR elements on both their chromosomes and

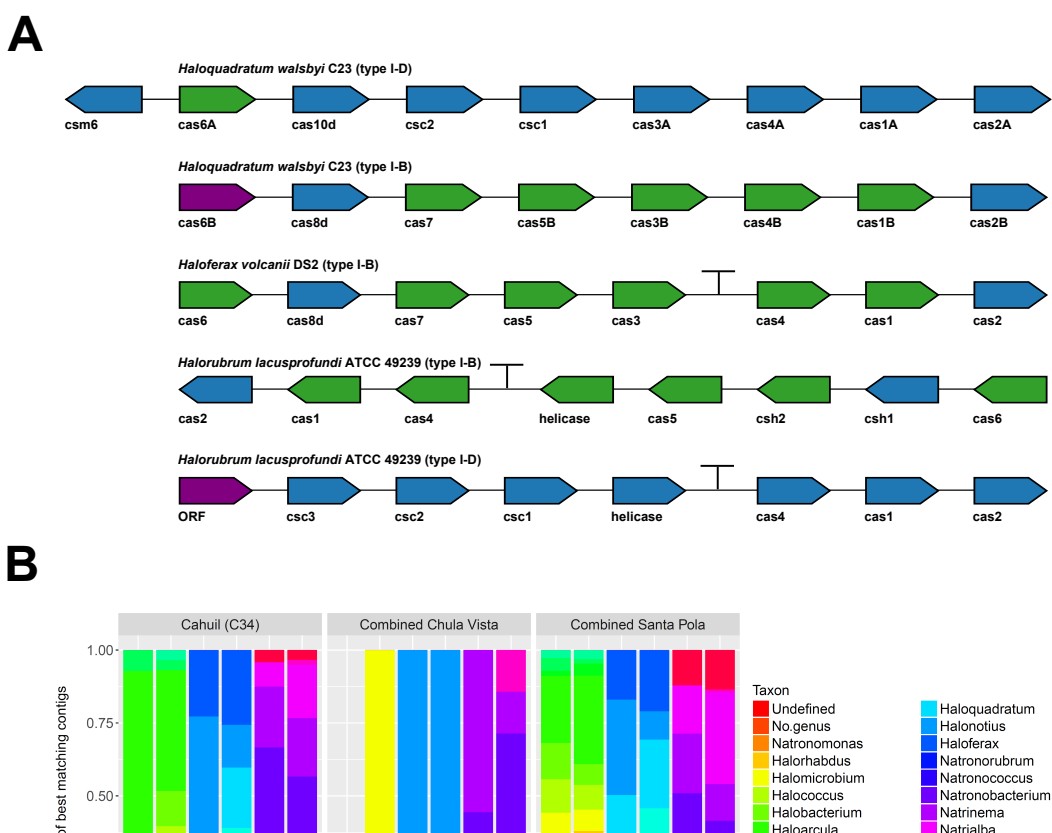

**Figure 10  Detection of *cas* operons in metagenomes and affiliation of detected genes to representative taxa.**  Three metagenomes (Cahuil/C34, combined Chula Vista, and combined Santa Pola and Isla Cristina) were assembled and searched for contigs matching *cas* genes. Detection results are reported for five halobacterial *cas* operons (A). *cas* genes detected in no metagenomes are shown in red; genes detected in one metagenome are shown in blue; genes detected in two metagenomes are shown in green (A). Detected contigs assembled with Newbler and Velvet were further aligned against a library of *cas* genes from three halobacterial orders (Halobacteriales, Haloferacales, and Natrialbales). The taxonomic affiliations of the best hits for each contig were tabulated into profiles for each order and assembly method (B).

plasmids, suggesting the possibility of horizontally transferred CRISPR systems (*Baliga et al., 2004*; *Grissa, Vergnaud & Pourcel, 2007*; *Hartman et al., 2010*; *Liu et al., 2011b*; *Han et al., 2012*). Viral interactions with *Salinibacter* remain far more elusive, as this bacterium lacks a CRISPR system (*Grissa, Vergnaud & Pourcel, 2007*).

Our CRISPR virus-host interaction studies help explain the paradox of high viral density and low infection rates in hypersaline ecosystems (*Brum et al., 2005*; *Bettarel et al., 2011*). Some studies have suggested high rates of lysogeny accounts for this paradox (*Bettarel et*

*al., 2011*), while others suggest saltern microbes have increased levels of viral resistance (*Brum et al., 2005*). Santa Pola saltern viruses have not been shown to hybridize to saltern microbes' genomes, excluding the lysogeny hypothesis (*Santos et al., 2010*). Our results suggest CRISPR defenses may have a role in reducing infection rates, as detected CRISPR spacers were mapped to known, sequenced saltern phages and CRISPR-associated (cas) genes were detected in studied saltern metagenomes, indicating saltern microbes do have CRISPR systems. Nonetheless, we still cannot exclude the possibility that innate phage resistance due to variation in cell surface proteins could also account for low viral infection rates (*Cuadros-Orellana et al., 2007*).

Nonetheless, the variation of CRISPR systems and resulting variation in *Haloquadratum* phage resistance could carefully control the archaeon's abundance. Together both haloviruses and CRISPR defense could tightly regulate the population of *Haloquadratum*, which is both the major component of many salterns and consumer of the glycerol metabolism intermediate dihydroxyacetone. Levels of CRISPR defense may thus account for the turnover of this organism. In combination with our studies of glycerol metabolism, our insights into virus-host interactions may also show how lytic viruses regulate glycerol turnover in these ecosystems. Our glycerol metabolism gene detection results suggest *Haloquadratum* and *Halorubrum* tend to oxidize and then phosphorylate glycerol, while other Halobacteriaceae and *Salinibacter* first phosphorylate and then oxidize glycerol. Knowing that previously identified Santa Pola haloviruses are lytic (*Santos et al., 2010*), viruses we mapped to *Haloquadratum* could either reduce overall rates of glycerol oxidation to dihydroxyacetone or release free dihydroxyacetone back into the ecosystem through lytic infection.

Future studies should further analyze the number and genomic arrangement of spacers in different *Haloquadratum* strains and examine how abundances of these populations change over time. Ideally, these studies would also relate changes in CRISPR arrays to changes in viral population structure over time. Such work would build upon recent studies of the temporal dynamics of virus-host interactions in Lake Tyrrell, an Australian hypersaline ecosystem (*Emerson et al., 2013*). Our studies and such future projects will help us better understand how viruses regulate hypersaline ecosystems across the planet.

## ACKNOWLEDGEMENTS

Thanks to Michael Crowder and Gary Lorigan (Miami University) for feedback on the project and manuscript.

### Funding

The project was funded partially by the Committee on Faculty Research (CFR) program, the Office for the Advancement of Research & Scholarship (OARS), and by an Academic Challenge grant from the Department of Biology (Miami University). The funders had no role in study design, data collection and analysis, decision to publish, or preparation of the manuscript.

### Grant Disclosures

The following grant information was disclosed by the authors:

Committee on Faculty Research (CFR) program.

Office for the Advancement of Research & Scholarship (OARS).

Academic Challenge grant from the Department of Biology (Miami University).

### Competing Interests

Chun Liang is an Academic Editor for PeerJ.

### Author Contributions

- Abraham G. Moller conceived and designed the experiments, performed the experiments, analyzed the data, contributed reagents/materials/analysis tools, wrote the paper, prepared figures and/or tables, reviewed drafts of the paper.
- Chun Liang conceived and designed the experiments, analyzed the data, wrote the paper, reviewed drafts of the paper.

### DNA Deposition

The following information was supplied regarding the deposition of DNA sequences:

The saltern metagenomes used are available at the NCBI Sequence Read Archive (SRA) via the following SRA accession numbers:

SRX328504, SRX024859, SRX352042, SRX347883, SRX090229, SRX680116

Additional metagenomes are also availabe at iMicrobe via the following web address: http://data.imicrobe.us/project/view/58.

### Data Availability

GitHub. MetaCRAST (reference-guided CRISPR detection): https://github.com/molleraj/MetaCRAST.

### Supplemental Information

Supplemental information for this article can be found online at http://dx.doi.org/10.7717/peerj.2844#supplemental-information.

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
