# Peer review of "Determining virus-host interactions and glycerol metabolism profiles in geographically diverse solar salterns with metagenomics"

_PeerJ, doi:10.7717/peerj.2844_

## Round 0.1 · original submission · Major Revisions

· Academic Editor

Major Revisions

Both reviewers are seriously concerned about the methodology in order to draw such conclusions as the title suggests. Reviewer 1 properly states that "The study overreacts on some of their conclusions and this could be due to excitment, but several times it is not supported by the data i.e. " ...we find that CRISPRs and viruses together help regulate glycerol consumption and dynamics in these hypersaline ecosystems..." or "Taken together, our results suggest that both viruses and CRISPR systems may regulate saltern ecology." Proper experiments would be needed in order to prove the assumptions, so it is necessary to modulate the conclusions reach" . Meanwhile reviewer 2 is very concerned about statistics. From my point of view each metagenome is unique and not a replicate of anything. Hence the authors should redress all those questions and have a more humble approach to what the data is really telling us. The reason I do not rejected the ms is because I believe that the story that the raw data can tell is interesting enough, that does not need to be amplified.

Reviewer 1 ·

Basic reporting

The manuscript by Moller and Liang study the CRISPR abundance in saltern metagenomic samples and they are presenting a new CRISPR de novo detection pipeline which is publicly available through Github. I think the paper shows comparative metagenomic results on the CRISPR abundance and how this could be connected to phage resistance in the microbial genomes.

There are several grammar flaws through the text and I would recommend to share the manuscript with colleagues and take note of the recommendations done here and probably from the other reviewers. Most concerning is the light use of adjectives which are merely anecdotal and most times do not show values or data support (most diverse, significant proportions, dominated, abundant component, much lower, etc.), please take note of this flaw and correct it.

There is some disorder in the manuscript structure where some methods goes into results and also in the opposite way. This needs to be corrected.

Experimental design

The strategy seems sound and the pipeline is also okay. What is needed is to explain the details clearly in the methods section.

The comparisons are interesting but is hard to follow up the subsets generated from each location and then which ones were assembled or not. Suggestions for this will be in the general comments to authors.

Validity of the findings

The study overreacts on some of their conclusions and this could be due to excitment, but several times it is not supported by the data i.e. " ...we find that CRISPRs and viruses together help regulate glycerol consumption and dynamics in these hypersaline ecosystems..." or "Taken together, our results suggest that both viruses and CRISPR systems may regulate saltern ecology." Proper experiments would be needed in order to prove the assumptions, so it is necessary to modulate the conclusions reach.

Another expample would be: "Taken together, this evidence strongly suggests that the Haloquadratum species in these salterns are undergoing net spacer loss." You would need to show that the ancestor Haloquadratum species host larger amounts of spacers to conclude this, and this is not shown anywhere in the paper.

While the assumptions could be logical they need to be moderated and disclaimed as suggestions or speculations.

Additional comments

L48. There is big debate about virus being lifeforms, please refer to them as entities
L77. alga -> algae
L129 Order the description, the metagenomic sets descriptions could be at the end of methods, you need to add the DB identifiers, you talk about assembly and CRISPR detection but its confusing to show this before describing how to perform this tasks.
L148 What is profile quality?
L184-189 Rewrite the MetaCrast pipeline description you are presenting it in this paper, please make a detailed description of the pipeline. The last two sentences of the paragraph are a total mess.

L197. necessitating?
L249 ID numbers?
L267. "Most diverse microbial communites" Please calculate alpha diversity metrics and state this with the current values.
Pages 15-18 REVIEW the use of your adjectives
L401-413 Discuss about the biological meaning of a small difference in nucleotide frequencies between virus and host.
L497-498 Vague idea, you need to mention briefly your evidences. What do you understand as saltern ecology?

Please review the attached annotated PDF on notes about your figures, general comments, missing references. Take special attention to the figures' footnotes which needs some work.

Annotated reviews are not available for download in order to protect the identity of reviewers who chose to remain anonymous.

·

Basic reporting

Please see "General Comments for the Author".

Experimental design

Please see "General Comments for the Author".

Validity of the findings

Please see "General Comments for the Author".

Additional comments

Major comments:
This manuscript was very challenging to review for 2 main reasons: (i) I am unsure if these many metagenomes are comparable quantitatively, and (ii) there are no statistical analyses performed to describe trends or comparisons reported in this manuscript. These issues are of critical importance, as the authors describe their results in the context of “higher”, “clustering”, “increased”, etc., yet there are no statistical analyses conducted to evaluate these observed results. I am wondering if the statistical analyses were not performed because the various datasets are not able to be compared *quantitatively* due to differences in metagenome generation methods (it is indicated in the manuscript that this is perhaps the case; i.e., differing read lengths between metagenomes). If that is the case, then the results comparing the metagenomes (most of the manuscript) are in question and the manuscript should be re-written as a *qualitative* analysis. If the authors feel that they can *quantitatively* compare these metagenomic results, then there needs to be an inclusion of statistical analyses to evaluate every statement where comparisons are made (i.e., “higher”, “clustering”, “increased”, etc.). To properly evaluate this manuscript, and the conclusions presented in the Abstract, I need to know whether this is a quantitative-based study, and if so, I need to see statistical analyses conducted to validate the authors’ statements. Because of these issues, I stopped reviewing the manuscript at Line 367, as I knew I could not evaluate the study as it is presently written and presented. To be clear, I feel that this study could be interesting and reveal important information; I just cannot review it accurately in its present form.

Minor Comments (please note that I stopped reviewing the manuscript at line 367):

Line 128: The methods section is very complex with many different bioinformatic tools used for different comparisons. I found myself drawing a diagram to keep track of all of the steps used to generate the results. Could the authors include such a diagram as a supplementary figure that will help the reader understand the overall process? (i.e., a flowchart showing the steps of data analysis and the resulting datasets that were used to interpret the data).

Lines 140-141: Upon initial reading, it seemed as though the authors had not stated the methods used to assemble the metagenomes. These methods are stated later in the manuscript (lines 223-224). I suggest either inserting a “see methods below” phrase, or removing this statement about assemblies from the paragraph since the paragraph is in regards to what metagenomes were used and where they came from.

Line 143: In this paragraph, please specify that the authors are examining microbial metagenomes (I assume these are not the viral metagenomes). Also it would be good to mention whether all of the microbial metagenomes were generated with the same methods (i.e., same prefilter size?, same sequencing platform, read length, and depth of sequencing?). This information will help the reader understand how comparable these diverse metagenomes are – please discuss this a bit as well for the readers’ benefit.

Lines 213-217: Is there a reference for this method? If so, please include it. If this is a method developed by the authors, please include information validating that it is accurate.

Lines 251-263: This discussion of prior results seems to belong in the Discussion section, as it seems to not include any results from the present study.

Line 267: I do not think the term “diverse” should be used here, as a diversity metric was not calculated. Further, the data presented in Fig 1A includes several different levels of taxonomy (i.e., phylum, order, etc.), and includes an “other” category; as such, it is difficult to gauge a diversity level from just looking at the graph. The authors should either calculate a diversity metric, or use a more appropriate description of the data (i.e., number of OTUs).

Lines 278-281: Please use a statistical analysis to evaluate the clustering of these samples in the PCA plot (i.e., are these clusters statistically significant?). For example, this can be done using factors in the program R.

Lines 300-304: This information also seems to belong in the Discussion section.

Lines 305-316: These trends with salinity need to be evaluated statistically (i.e., is there a significant trend with salinity? Are these values significantly different between salterns?).

Figure 2: Please indicate the units for the y-axis. Please also define the abbreviations for each gene for ease of reading this figure.

Figure 3: Please indicate the units for colors on the heatmaps.

Lines 320-341: It is difficult to determine whether the described “clustering” is significant or not. Please include some statistical analyses to validate the results stated here.

Lines 351-353: Again, please conduct analyses to state whether these differences are statistically significant or not.

Figure 4: For this data to be comparable, the number of CRISPRs detected should be reported as relative abundance normalized to the metagenome size (i.e., level of sequencing). This is shown in Figure S3, but that information should be in the main text because it is the informative data – the total number of CRISPRs is not informative as a comparable metric.

Lines 363-367: Please include quantitative results here instead of the general terms “a number of”, “higher”, “some”.

---

## Round 0.2 · Major Revisions

· Academic Editor

Major Revisions

Dear Dr. Liang,

Thank you for your Appeal. As you know, the prior Editor declined to handle your Appeal. As a result I have been asked to consider it.

I am returning the submission to you so that you can upload the updated rebuttal and the document with tracked changes which you referenced in your earlier correspondence.

I will then proceed with the evaluation of your Appeal.

Thank you very much for your patience.

· Appeal

Appeal

For our manuscript "Determining virus-host interactions and glycerol metabolism profiles in geographically diverse solar salterns with metagenomics" (#2016:06:11149:1:0:REVIEW), we are very surprised by your decision that "The authors did not correct the major issues that worried both reviews and did a very shallow correction on the paper. I consider that to take lightly the opinions of experts is not a good message. the authors did not readdress any of the concern points". This is not the case for our revised manuscript because we did take great efforts to address every comment or concern from two reviewers.

I am afraid that the presentation of our rebuttal letter (actually it should be called point-to-point response letter) gave you a wrong impression. I apologize for such bad presentation of our letter. Here in this email, you can find our modified letter that provides clear description about what we have done in the revision, together with the revised manuscript in which all changes are highlighted in red font. The manuscript in attachment is the same as the one that we have submitted to PeerJ for revision except all changes in the submitted one are tracked using Microsoft "track changes" function.


· · Academic Editor

Reject

The authors did not correct the major issues that worried both reviews and did a very shallow correction on the paper. I consider that to take lightly the opinions of experts is not a good message. the authors did not readdress any of the concern points.

---

## Round 0.3 · Major Revisions

· Academic Editor

Major Revisions

Dear Abraham and Chun,

I apologize for the lengthy editorial process with respect to your work. I have been waiting for the second reviewer to chime in, however, thanks to the extraordinary effort Jennifer Brum put into evaluating your revised manuscript, I feel comfortable moving forward without delaying this process any longer.

As you will find evidence in her evaluation, one of the outstanding concerns Dr. Brum still has is the disconnect between the Results and Discussion sections of your manuscript. I would like to encourage you consider carefully merging the two into a single "Results and Discussion" section to clear up some of the disconnected parts.

As you will see in the detailed evaluation of Dr. Brum, there are some other remaining issues. Although as far I as I can see there aren't any significant challenges to address these concerns, here I am sending a "Major Revision" decision, simply for logistical reasons. But I would like to mention that thanks to your revised rebuttal, I think the improvement of the manuscript compared to the initial version is quite clear.

I would also like to thank Jennifer in this letter for taking the time to go through your revisions, and further improve this study with her suggestions. It is absolutely a humbling experience to witness such meaningful exchanges between peers.

Thank you in advance for your attention to these suggestions, and for your patience with the long editorial process.

·

Basic reporting

Please see General Comments.

Experimental design

Please see General Comments.

Validity of the findings

Please see General Comments.

Additional comments

I appreciate the changes and improvements the authors have made to this manuscript to make it easier to understand, as well as their inclusion of statistical analyses to quantitatively analyze the data. I still have some rather substantial comments regarding additional/different statistical analyses and such (which should be easily addressed by the authors), but my main concern is in regards to improving the Discussion section and tying it to the Results (please see comments at the end of my review).

Lines 35-36: The characteristics and names of the salters have not yet been described at this point in the manuscript, so inclusion of the saltern abbreviations in the abstract is not yet understandable to the reader. Can the authors simplify this sentence and state that there were more CRISPR spacers detected in the Haloquadratum-dominated salterns? (the use of the word “detected” is needed because this is not quantitative data)

Lines 92-93: I do not think the phrases “whole metagenome sequencing” and “total microbial DNA” are appropriate here because you are never actually sequencing the whole metagenome or all of the DNA, only a portion of it. I suggest revising to “metagenomic sequencing” and “microbial community DNA”, respectively.

Tables 1 and 2: These are the first tables or figures cited within the manuscript, yet they reference figures to define the abbreviations used for the saltern names. These tables would thus be the appropriate place to include the full saltern names and their locations as they are the first time these abbreviations are used.

Lines 159-160: Please include citations for R and ADONIS, as well as the R version number.

Line 188: Somewhere in the Methods description of CRISPR analyses, there should be a statement that these are treated as qualitative analyses and a description of why they are qualitative. This would be the same explanation as the authors provided to me in their Response to Reviewers, and will help the reader understand how these analyses are used.

Lines 205-218: Please check the tense in this section. It seems the authors have switched to present tense instead of the past tense used for Methods sections.

Line 234: Figures should be introduced in order. In this case, mentioning Figure 4 in the sentence is not necessary to describe the method used, and it can be removed.

Line 255: Please include the reference for the MATLAB Bioinformatics Toolbox.

Lines 315-321: Please report whether salinity was a significant factor in structuring the similarities/differences between these samples. This can be done (for example) with response surfaces using the orsidurf function in vegan in R (I do not know whether Adonis can do this as well).

Line 330: In the Introduction (lines 78-81) it is stated that glycerol “may be” the main source of carbon in salterns, yet here the authors state that glycerol “is” the main source of carbon. Please adjust these sentences for continuity.

Line 344: In the text, the authors refer to “correlations”, however in Figure 2 and in the text (lines 345-349) the authors report regression results. Correlations and regressions are not identical, and correlations should be used here due to the fact that the authors are relating 2 independent variables (regressions are used for one dependent and one independent variable). For correlations, please check for normal distribution of the data if using parametric statistics (i.e., Pearson correlation), then report the correlation coefficient and p-value.

Figure 2: First, the graphs in this figure include different numbers of data points. Why is this the case?? I am guessing that any metagenome with no gene detected was removed from the analysis, which is not correct. These figures and statistical analyses *must* include all data, even if there is zero abundance of that gene. Second, the graphs in this figure should not include negative values on the axes, as it is impossible to have negative abundances of genes. Third, please report p-values for these relationships. Fourth, please report the units for both the x and y axes. Fifth, please adjust all y-axis values to have the same format – it appears that an exponent format (as in the central graph) would be most appropriate given the number of decimal places required. Sixth, the Dak_2 graph lacks values on the x-axis. Seventh, I do not think it is appropriate to include an R-squared value for a relationship between 2 data points (i.e., Dak_2) – although this will be corrected when the authors include all metagenomes where there are zero Dak_2 genes detected.

Figure 3: In the caption it is stated that part C has hierarchical clustering, however I do not see clustering in the graph – please correct this. Please also define “level” in part C, as I do not know what that means here.

Lines 397-398: As I stated previously, there should be some statement here regarding the qualitative nature of these data. This is necessary for the reader to understand that the results are not quantitative, and the reason why they are not quantitative (again, use the same explanation you provided for me in the Response to Reviewers).

Figures 7 and 8: I find these figures to be very difficult to understand because they are entirely made up of abbreviated sequence IDs with no indication of what they mean (i.e., any reader will not know what G467… means). My suggestion is to include the information available about the taxonomy (if available) of these sequences within the figure itself – if the taxonomy is not available, then insert “unknown”. The sequence ID can be included in parentheses. This information is partially included within the text (i.e., lines 417-421), and that is the information that is most relevant for the reader (i.e., that G467 was affiliated with Haloquadratum; that G85 was affiliated with unknown hosts). These figures should stand on their own and not require a reader to check both the text and tables to find out what each sequence ID means.

Figure 9: In the figure there is data described as “dimer usage” and “codon usage”, yet in the text they are referred to as “dimer usage” and “trimer usage” (line 447). Please use the same phrases in the text and figure.

Lines 557 and 560: The term “viral predation” seems a bit overreaching here because the authors have not evaluated the level of mortality associated with viral infections. The authors are only connecting viruses with hosts based on CRISPR-based and nucleotide-based analyses and cannot state anything about the level of mortality from this data.

General Discussion Comment 1: I find the Discussion to be a bit difficult to interpret for a few reasons: (i) There are no figure references included to tell me what data the authors are describing in each section; (ii) There are many sentences that discuss the data currently presented in the manuscript, but with references at the end of the sentence such that I cannot determine what information is from the present study and what information is from prior studies (e.g., lines 481-483), thus I suggest rephrasing much of the discussion in the format of “we found this, which was in agreement/connected with prior studies (cite reference)”; (iii) there are no subheadings as in the results section and this also makes it difficult to associate the results with the discussion. Based on this, I strongly suggest merging the Results with the Discussion, followed by a Conclusions section (if this is allowed in PeerJ). This will help the reader tie together the results with the authors’ interpretation of the results.

General Discussion Comment 2: Throughout the manuscript I was struck by the seemingly disparate topics covered by the research (i.e., glycerol metabolism and virus-host interactions). This crossover is mentioned in the Introduction (lines 120-122) and only in one sentence in the Discussion that I can see (lines 562-563). I feel that there should be much further discussion about this connection – i.e., I am still struggling to understand the reasoning that connects (1) glycerol metabolism genes affiliated with specific microbes to (2) virus-host interactions investigated using the CRISPR defense mechanism which is not at all the only defense mechanism available in virus-host interactions. I am sure that the authors understand this association of topics, but it is not clearly laid out in the manuscript for the reader, so please “spoon-feed” this concept to the reader and increase the discussion and interpretation of these relationships between the topics.

General Discussion Comment 3: As a researcher who has studied a moderately hypersaline environment, I am still perplexed by the paradox that hypersaline environments have the highest abundances of viruses and microbes (and thus the highest contact rates), yet very low frequency of visibly infected cells. This has been noted by several studies including my own (Brum et al 2005, Aquatic Microbial Ecology; Guixa-Boixareu et al. 1996, Aquatic Microbial Ecology; and I think Bettarel et al 2011, FEMS Microbiology Ecology). One of the hypotheses to explain this is that there may be extraordinarily high diversity in these systems, yet the authors state that diversity is low in the salterns (lines 62-64), so that does not seem to be the reason. Another hypothesis is that there may be greater lysogeny at higher salinities (Bettarel et al 2011). But finally, the third hypothesis is that there is increased viral resistance in microbes in hypersaline environments (as I suggest in Brum et al 2005, but with no data at all to investigate this idea). So, can the authors use the data in the present manuscript to shed light on this paradox? i.e., is there any evidence of greater resistance mechanisms (here just the CRISPR system) in hypersaline environments to explain low infection frequencies with such high perceived contact rates between viruses and hosts? I know this requires an additional paragraph or two from the authors, but I really think addressing this issue would help put the authors’ research more firmly within the context of hypersaline viral research, and thus elevate this manuscript to address a greater conceptual problem, even if the authors do not have the quantitative data to conclusively solve this paradox.

---

## Round 0.4 · accepted · Accept

· Academic Editor

Accept

I am happy to let you know that your revisions were satisfactory, and I believe your work is ready to be published. Thank you very much for your patience, and your effort to address all reviewer concerns.

·

Basic reporting

No Comments.

Experimental design

No Comments.

Validity of the findings

No Comments.

Additional comments

I have no further comments and suggest that this revised version be accepted as is.